# Apoptosis and Pharmacological Therapies for Targeting Thereof for Cancer Therapeutics

**Vishakha Singh** [1], **Amit Khurana** [2,3,4,*], **Umashanker Navik** [5], **Prince Allawadhi** [6], **Kala Kumar Bharani** [4,*] **and Ralf Weiskirchen** [2,*]

1   Department of Biosciences and Bioengineering, Indian Institute of Technology (IIT), Roorkee 247667, India; 93vishakhasingh@gmail.com

2   Institute of Molecular Pathobiochemistry, Experimental Gene Therapy and Clinical Chemistry (IFMPEGKC), RWTH Aachen University Hospital, Pauwelsstr. 30, D-52074 Aachen, Germany

3   Department of Veterinary Pharmacology and Toxicology, College of Veterinary Science (CVSc), Rajendranagar, Hyderabad 500030, India

4   Department of Veterinary Pharmacology and Toxicology, College of Veterinary Science (CVSc), Mamnoor, Warangal 506166, India

5   Department of Pharmacology, Central University of Punjab, Ghudda, Bathinda 151401, India; usnavik@gmail.com

6   Department of Pharmacy, Vaish Institute of Pharmaceutical Education and Research (VIPER), Pandit Bhagwat Dayal Sharma University of Health Sciences (Pt. B. D. S. UHS), Rohtak 124001, India; princeallawadhi@yahoo.com

\*   Correspondence: ak3.khurana@yahoo.com (A.K.); bkalakumar@gmail.com (K.K.B.); rweiskirchen@ukaachen.de (R.W.)

**Abstract:** Apoptosis is an evolutionarily conserved sequential process of cell death to maintain a homeostatic balance between cell formation and cell death. It is a vital process for normal eukaryotic development as it contributes to the renewal of cells and tissues. Further, it plays a crucial role in the elimination of unnecessary cells through phagocytosis and prevents undesirable immune responses. Apoptosis is regulated by a complex signaling mechanism, which is driven by interactions among several protein families such as caspases, inhibitors of apoptosis proteins, B-cell lymphoma 2 (BCL-2) family proteins, and several other proteases such as perforins and granzyme. The signaling pathway consists of both pro-apoptotic and pro-survival members, which stabilize the selection of cellular survival or death. However, any aberration in this pathway can lead to abnormal cell proliferation, ultimately leading to the development of cancer, autoimmune disorders, etc. This review aims to elaborate on apoptotic signaling pathways and mechanisms, interacting members involved in signaling, and how apoptosis is associated with carcinogenesis, along with insights into targeting apoptosis for disease resolution.

**Keywords:** apoptosis; carcinogenesis; BCL-2; signaling; therapy; caspase; extrinsic pathway; intrinsic pathway; inhibitors; tumor suppressor

## 1. Introduction

The term "Apoptosis" originates from Greek, which means the shedding of leaves from trees in autumn or the falling of petals from flowers. Apoptosis was firstly utilized by Kerr, Wyllie, and Currie in 1972 for explaining a morphologically discrete way of cell death. However, multiple concepts of apoptosis were precisely explained several years back [1–3]. The main concept of apoptosis emerged from the knowledge of the process of programmed cell death that takes place in the developmental cycle of *Caenorhabditis elegans*. There are 1090 somatic cells that are produced in the adult worm; out of these, 131 cells go through the pathway of programmed cell death or apoptosis during different stages of the developmental cycle [4–6]. It became clear that apoptosis is a recognized and discrete way of cell death that allows programmed cell death or elimination of genetically determined

cells. However, some other ways of programmed cell death are also present physiologically such as autophagy and necrosis. In this review, we majorly focus on apoptosis [7,8].

Apoptosis is an orderly orchestrated process that takes place in the different stages of physiological and pathological processes. It is distinguished by events of remarkable perturbations to the cellular framework, which involve not only cell death, but also encourage the elimination of cells by phagocytes and impede unfavorable immune responses [9]. The process of apoptosis is operated by a family of caspases. They attack cellular proteins for regulated proteolysis and also prevent neighboring cells from damage caused by programmed cell death by controlling the discharge of immunostimulatory molecules [10]. Apoptosis plays an important role in the survival of an individual and is appraised as a crucial component in several cellular processes, such as normal embryonic growth and development, maintaining cellular turnover, normal growth and development, immune functioning, proliferation of mutated chromosomes, hormone-mediated atrophy, and eradication of indisposed cells and in cell homeostasis [7]. Apoptosis takes place naturally during growth and developmental stages, aging and in maintaining homeostasis for definite cell populations [7]. Moreover, apoptosis can also occur as a defense operation in case of disease, such as pathogenic infections or during immune reactions, as shown in Figure 1. However, apoptosis is controlled and modulated by several stimuli and factors, but all cells do not respond to the same stimulus [11–15].

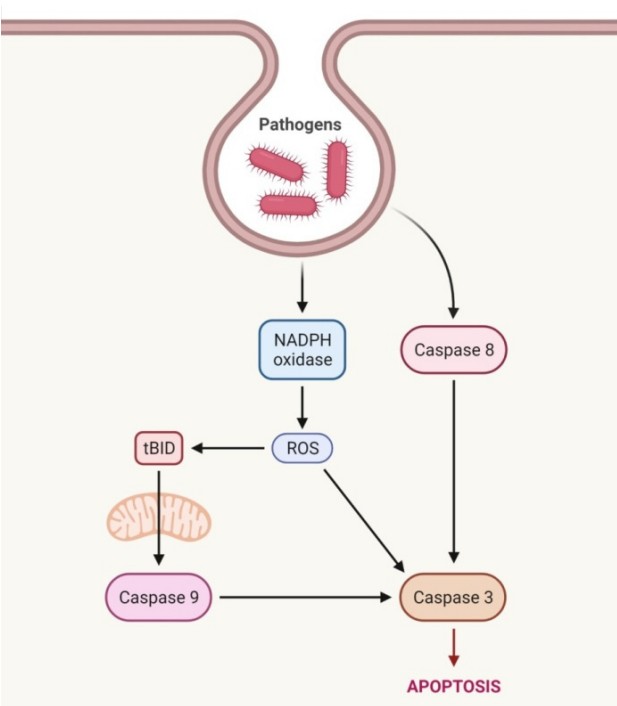

**Figure 1.** Apoptosis during pathogenic infection. During pathogenic infection, apoptosis is promoted by enhanced expression of B-cell lymphoma 2 (BCL-2) family proteins such as BH3-interacting domain death agonist (BID) and its truncated form (tBID), along with increment in formation of reactive oxygen species (ROS). The figure was created with BioRender [16].

There are numerous stimuli and situations that can activate apoptosis, but all cells do not die in response to the same stimuli [17,18]. These stimuli could be radiation or drugs, which are utilized in cancer therapy, leading to DNA disintegration in some cells [19]. Additionally, glucocorticoids are known to trigger the apoptosis in lymphocytes, and synthetic glucocorticoids are given for the treatment of hematological malignancies [20,21]. Apoptosis has extensive biological importance. It is necessary for cellular physiological processes in eukaryotes, such as tissue homeostasis, characterization of immune cells, tissue formation, nervous system, and in carcinogenesis [22]. In the central nervous system,

apoptosis is linked with several disorders such as in Alzheimer's disease, Parkinson's disease, lateral sclerosis, Huntington's disease and other debilitating diseases [23–25]. Apoptosis is associated with removal of damaged, unwanted cells, cells with non-repairable DNA and auto-reactive cells in immune system [26]. Hence, it is not surprising that dysfunction in apoptosis can cause cancer and autoimmune diseases [27,28]. Additionally, some evidence shows that dysregulation of apoptotic pathways leads to the development of several diseases, such as ischemic diseases, acquired immune deficiency syndrome (AIDS), and neurodegenerative disorders [7]. Therefore, the present review aims to highlight the importance of apoptotic signaling routes and mechanisms, as well as the interacting components involved in signaling, and how apoptosis is linked to carcinogenesis, as well as insights into using apoptosis to treat disease. Furthermore, we discuss the several targets and the potential lead compounds under investigation for the treatment of cancer.

## 2. Process of Apoptosis

The biology of apoptosis is complicated and includes an energy driven process of molecular episodes. There are two apoptotic pathways, namely, the extrinsic pathway, which is also named as death receptor pathway, while the other is intrinsic pathway, which is named as a mitochondrial pathway. These pathways are associated with each other and influence the functioning of each other, as shown in Figure 2 [29]. Additionally, there is one more pathway through which apoptosis is induced, which includes T-cell-mediated cytotoxicity and perforin/granzyme-mediated cell death. All the three pathways combine together at the execution pathway [30]. The execution pathway includes the cleavage of caspase-3 and finally leads to disintegration of DNA and cytoskeletal as well as nuclear proteins. It also involves the generation of apoptotic bodies, expresses ligands for receptors of phagocytic cells and is consequently engulfed by phagocytic cells [7].

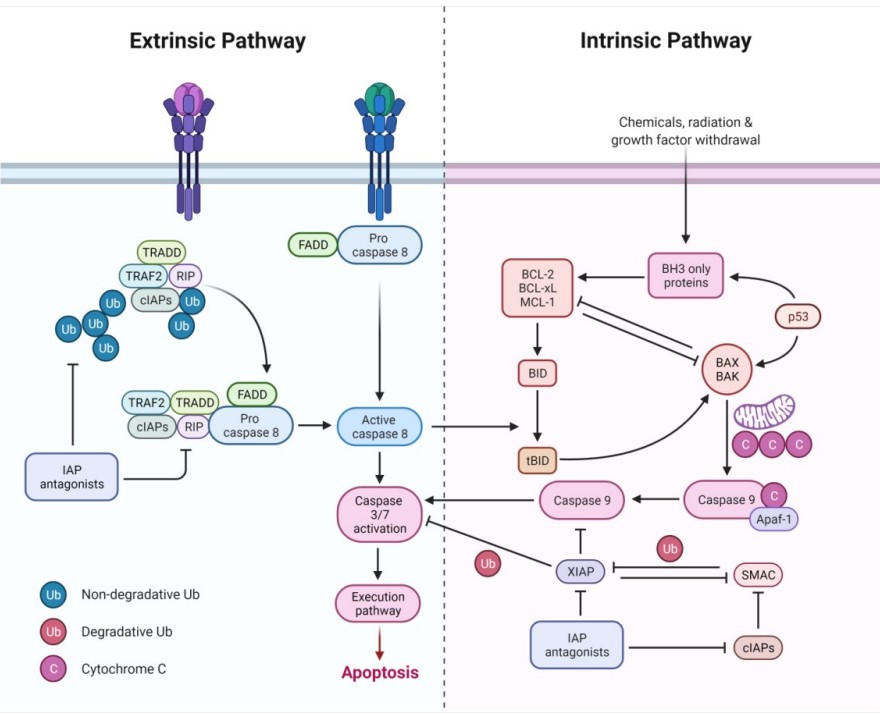

**Figure 2.** Apoptosis is mediated via extrinsic and intrinsic pathways. To begin an energy-driven process of molecular episodes, each pathway requires its unique triggering signal. Both routes activate their own initiator, which is accompanied by caspase-3 activation. Other protein families, such as inhibitors of apoptosis proteins (IAPs) and B-cell lymphoma 2 (BCL-2) family proteins, are also required for the entire mechanism. The figure was created with BioRender [16].

### 2.1. Extrinsic Pathway

The extrinsic signaling is mainly linked with transmembrane receptor-mediated interactions, which include death receptors of the tumor necrosis factor (TNF) receptor gene superfamily. The death domain of death receptors is responsible for transmitting death signals from the cell's surface to the intracellular signaling pathways [31]. There are different death receptors, which are well characterized, such as TNF-$\alpha$/TNFR1, FasL/FasR, apoptosis antigen (APO) 2L/DR4 APO3L/DR3, and APO2L/DR5 [32]. The mechanism of extrinsic pathway is well explained by the FasL/FasR and TNF-$\alpha$/TNFR1 models [33]. The Fas and TNF ligands bind to the Fas and TNF receptors, respectively, which in turn allow the interaction with their respective adapter proteins, namely Fas-associated protein with death domain (FADD) and TNFR1-associated death domain protein (TRADD) [34]. The binding of the adapter protein is linked with stimulation of procaspase-8 accompanied by the autocatalytic action of caspase-8 [32]. Once caspase-8 is stimulated, this pathway converges with the execution pathway.

Importantly, the apoptosis ligand 2/tumor necrosis factor-related apoptosis-inducing ligand (APO2L/TRAIL) pathway plays an important role in the progression and development of cancer and it acts independently of p53. Further, pro-apoptotic receptor agonists (PARAs) that target DR4 and/or DR5 have potential to give significant clinical benefit by killing tumor cells that are resistant to traditional chemotherapeutic agents [35]. Hence, targeting extrinsic pathways could be a novel treatment strategy to promote apoptosis in the cancer cells.

### 2.2. Perforin/Granzyme Pathway

In this pathway, T-cell cytotoxicity is mediated by cytotoxic T lymphocytes (CTLs), which are enrolled in targeting the cells for killing through the extrinsic pathway, as shown in Figure 3 [36,37]. They show cytotoxic effects by secreting perforin and releasing cytoplasmic granules, which involve serine proteases granzyme A and granzyme B in complex with serglycin [38]. Granzyme B can activate procaspase-10 and is able to activate caspase-3 directly in such a way that bypasses the upstream signaling cascade and directly stimulates the execution pathway [39]. Granzyme A is also associated with CTL-mediated apoptosis in a caspase-independent manner. It stimulates DNA nicking leading to the degradation of apoptotic DNA [7].

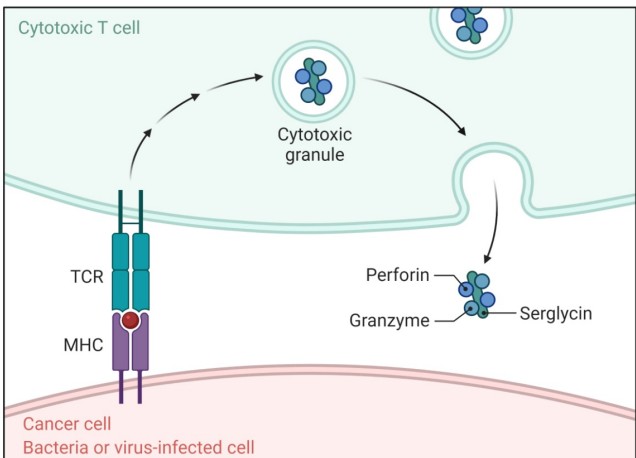

**Figure 3.** Function of the T cell receptor (TCR) and peptide–MHC complex in apoptosis. The TCR and peptide–MHC complex causes directed release of perforin and granzymes complexed with serglycin triggering apoptosis.

### 2.3. Intrinsic Pathway

The intrinsic pathway induces apoptosis via varied types of non-receptor-mediated stimuli, which are mainly mitochondrial-dependent events [40]. These stimuli generate intracellular signals that target the cell directly either in a positive or negative manner [7]. Positive stimuli include viral infections, radiation, hypoxia, toxins, hyperthermia and free radicals. Negative stimuli involve a lack of some growth factors, cytokines or hormones, which subsequently results in the repression of the death program, in turn, stimulating apoptosis. These mentioned stimuli bring about modifications in the inner mitochondrial membrane resulting in the liberation of pro-apoptotic proteins. One group of pro-apoptotic proteins consists of SMAC/direct IAP binding protein with low pI (DIABLO), cytochrome c and HTRA2/OMI, which activate the caspase-associated with the mitochondrial pathway [41]. Among these proteins, SMAC/DIABLO and HTRA serine peptidase 2 (HTRA2)/OMI stimulate apoptosis by preventing the functioning of IAP [41,42]. However, cytochrome c works by stimulating apoptotic protease-activating factor 1 (APAF-1) and procaspase-9 by binding to them, thereby resulting in the activation of caspase-9 [43,44]. Endonuclease G and apoptosis-inducing factor (AIF) are nucleases located in the intermembrane space of the mitochondria that together with caspase-activated DNase (CAD) lead to chromatin condensation and DNA fragmentation at later stages of apoptosis. Moreover, the members of the BCL-2 family are crucial in maintaining mitochondrial dependent apoptotic events as this family of protein regulates the permeability of the mitochondrial membrane and, hence, can act as pro-apoptotic or anti-apoptotic [45]. The pro-apoptotic genes include BAX, BCL-10, BIK, BAK, BLK, BAD, BIM, BID, PUMA and NOXA, while the anti-apoptotic genes include BCL-2, BAG, BCL-X$_S$, BCL-X$_L$, BCL-x and BCL-w. All these proteins decide the fate of cell, i.e., whether the cell undergoes apoptosis or aborts apoptosis [45–47]. Hence, modulating the expression of these proteins can regulate the apoptosis process and, thus, may halt cell proliferation in cancer cells.

### 2.4. Execution Pathway

The execution pathway connects both the intrinsic and extrinsic pathway and finally leads to apoptosis. In this cascade, the execution caspases activate and start the final process of apoptosis. Herein, activation of endonucleases and nucleases for the disintegration of nuclear material and cytoskeletal proteins is evident [7]. The main caspases of the execution pathway are caspase-3, caspase-6 and caspase-7, which degrade cytokeratins, poly (ADP-ribose) polymerase (PARP), cytoskeletal protein, namely, α-fodrin, the nuclear protein nuclear-mitotic apparatus protein (NuMA) and others, finally leading to several biochemical and morphological alterations in cells undergoing apoptosis [48]. For instance, caspase-3 is significant for cell fate following apoptosis and it activates CAD. In a normal cell, CAD remains in a binding state with its inhibitor, namely, the inhibitor of caspase-activated DNAse (ICAD), while in apoptotic cells, caspase-3 degrades ICAD, thereby releasing CAD [49]. Finally, CAD performs its function of fragmenting chromosomal DNA, resulting in the condensation of chromatin [49]. It also stimulates the process of cytoskeleton organization and cell fragmentation into apoptotic bodies.

Overall, this section discussed the involvement of different apoptosis pathways, such as the extrinsic pathway, the intrinsic pathway, and T-cell mediated and perforin/granzyme-mediated cytotoxicity, which ultimately combine with the execution pathway. Further, aberrant apoptosis is an important reason for resistance to cancer therapeutics. Additionally, cancerous cells can lead to altered expressions of several apoptotic and anti-apoptotic proteins, which ultimately may lead to increased cell proliferation. Hence, targeting these proteins opens up new possibilities for selectively eradicating cancer cells.

## 3. Caspases

What causes the biochemical and morphological changes linked with the process we recognize as apoptosis? The answer is caspases. The operation of apoptosis is driven by the activity of the family of cysteine proteases, which specifically make cuts at their substrate,

i.e., aspartic acid residues, and are called as caspases [50]. The name caspase is given for Cysteine Aspartyl-specific Proteases. In all animal cells, these cysteine proteases are available in the form of inactive zymogens, but become stimulated to acquire their active state and perform proteolytic processing specifically at aspartate residue [51]. Caspases are crucial in regulating the process of inflammation and cell death. The association of caspases in cell death was known long after the discovery of *ced-3* in *Caenorhabditis elegans*. Functionally, the main apoptotic caspases in mammals are caspase-2, -3, -7, -8, -9 and -10, while the inflammatory caspases are -1, -4, -5, -11 and -12. On the basis of a gene sequence arrangement, i.e., the existence or lack of protein interaction domains at the N-terminus, these apoptotic caspases are categorized as initiator caspases and effector caspases [52,53]. The initiator caspases consist of either death effector domain (DED domain) or caspase-recruitment domain (CARD domain). The DED domain is included in caspase-8 and -10, while caspase-2, -9, -1 and -11 contain CARD domains. These caspases regulate the process of activation of apoptosis. The functioning of initiator caspases is driven by the mitochondrial or intrinsic pathway and the extrinsic pathway [54]. As discussed earlier, the intrinsic pathway works in response to stress stimuli and is governed by the BCL-2 family of proteins leading to stimulation of BAX and BAK, which allows cytochrome c liberation and disturbs mitochondrial outer membrane permeability (MOMP) [55]. Cytochrome c binds with APAF-1 generating apoptosome and finally activates caspase-9. The extrinsic way of apoptosis is followed by ligand binding and via activating the death domain receptor family, such as TNFR, Fas and TNF-related apoptosis-inducing ligand (TRAIL). These allow activation of caspase-8 or -10 by generating the death-inducing signaling complex (DISC) [56]. As the activation caspases activate following their respective pathways, they undergo the activation of execution caspases, namely, caspases-3, -6 and -7. Caspase-2 is also crucial as it becomes stimulated via stimulus in both upstream and downstream of MOMP. Caspase-2 is linked to a large multiprotein complex known as PIDDosome, which consists of two proteins, namely RIP-associated ICH1/CED3-homologous protein with a death domain (RAIDD) and p53-induced protein with a death domain (PIDD). It can interact with these two proteins in combination as well as independently [57]. Overall, caspases play a crucial role in the initiation of apoptosis signals and induction of proteolysis, thereby causing cell death. Hence, targeting different caspases could be a crucial target for modulating the apoptotic machinery in cancerous tissue. Table 1 summarizes the role of caspases, their origin and their substrates.

**Table 1.** Caspases with their location, substrate and functions *.

| Caspases | Presence | Substrates | Functions |
|---|---|---|---|
| Caspase-1 | Spleen, liver, kidney, lung, heart | Lamins, Interleukins | Involved in inflammation, apoptosis induction when overexpressed [58] |
| Caspase-2 | Liver, CNS, kidney and lung development in embryo | Lamins, Golgin-160 | Apoptosis [59] |
| Caspase-3 | Broadly distributed | Caspases-6, -7, -9 | Apoptosis [60] |
| Caspase-4 | Lung, placenta, ovary, liver | Caspase-1 | Apoptosis [61] |
| Caspase-5 | Liver, lung | Max | Apoptosis, inflammation [62] |
| Caspase-6 | Liver, lung, skeletal muscle | PARP, caspase-3, NuMA, lamins, FAK, keratin-18 | Apoptosis [63] |
| Caspase-7 | Lung, kidney, liver, heart, spleen, testis | PARP, GAS2, EMAP II, calpastatin, FAK | Apoptosis [64] |

**Table 1.** *Cont.*

| Caspases | Presence | Substrates | Functions |
|---|---|---|---|
| Caspase-8 | Leukocytes, thymus, spleen, liver | Caspases-3, -4, -6, -7, -9, -10, -13 | Apoptosis [65] |
| Caspase-9 | Heart, liver, skeletal muscle, pancreas | Caspase-3, PARP, procaspase-9, caspase-7 | Apoptosis [66] |
| Caspase-10 | Tissues | Caspases-3, -4, -6, -7, -8, -9 | Apoptosis [67] |
| Caspase-11 | Brain microglia | Caspases-3, -1 | Apoptosis, inflammation [68] |
| Caspase-12 | Endoplasmic reticulum (ER) | Caspases-1, -4, -5, -11 | Apoptosis-mediated by ER stress [69] |
| Caspase-13 | Lymphocytes, placenta, spleen | Caspase-8 | Inflammation [69] |
| Caspase-14 | Epidermal cells | Caspases-8, -10 | Inflammation [70] |

\* Abbreviations used are: EMAP II, endothelial monocyte-activating polypeptide-II; FAK, focal adhesion kinase; GAS2, growth arrest-specific protein 2; Max, Myc-associated factor X; NuMA, nuclear mitotic apparatus protein; PARP, poly(ADP-ribose) polymerase.

## 4. Apoptosis-Associated Protein Domains

There are proteins or families of proteins that regulate the caspase activation pathways, namely, the extrinsic or intrinsic pathways, and they are identified depending on their amino acid sequence or homologue. The interactions facilitated by these protein families are driven through protein domains that are linked with the regulation of apoptosis, such as death domains (DDs), caspase recruitment domains (CARDs), death effector domains (DEDs), BCL-2 family proteins and of IAP-family proteins.

### 4.1. Death Domain Proteins

DDs consist of a condensed bundle of six alpha helices that interact among themselves and form an oligomer. The differences in surface residue decide the specificity for partner selection in the death domain [71]. Several TNF family members of the cytokine receptor have their death domains in the cytosolic face. The TNF receptor family, including TNFR1, DR3 and DR6, works by binding to adapter protein, namely, TRADD, which contains its homologous death domain. The DD of TRADD is able to bind with other proteins containing DD, such as Fadd, an adapter protein. The FADD protein has a DD through which it associates with the TNF family and, hence, links TNF family receptors to caspases [72]. Hence, this protein plays a role of mediator. Additionally, a similar example is seen in case of RAIDD, which is also known as CRADD. This is an adapter protein containing DD along with CARDs. It aids in linking the death receptor family with procaspases by binding with the CARD domain of pro-caspase-2 [73]. Moreover, TRAIL-R1 (DR4) and TRAIL-R2 (DR5) are involved in the induction of apoptosis by binding to the TRAIL ligand. TRAIL ligand can also bind with DcR1, DcR2 and osteoprotegerin (OPG), which are also TNF family proteins, but they are considered as decoy receptors as they do not carry a death signal [74]. Fas, is also a member of TNF receptor family and is considered a potential stimulator of apoptosis. It is crucial in the homeostasis of the immune system, removing autoreactive lymphocytes and reducing immune response once foreign antigens are removed [75,76]. There are some other proteins that have DD and are involved in apoptosis, such as DAP kinase, but their mechanism is not well understood. However, it has been reported that they activate caspase-8, which disturbs the cytoskeleton [54]. Due to the importance of DD proteins, any imbalance or defect in the regulation and functioning of these proteins leads to human diseases. For instance, upregulation in the expression of FasR or FasL on lymphocytes has been known to be associated with HIV infection. Moreover, mutation in death domain of the Fas gene can cause lymphoproliferative syndrome, an autoimmune syndrome and malignancies [77].

### 4.2. Death Effector Domain Proteins

The death effector domain (DED) shares similarity with DD proteins in terms of structure. It exists in initiator caspases, namely, caspase-8 and caspase-10. There are two tandem DED motifs present in the prodomain region of initiator caspases. Due to the presence of DED in caspase-8 and caspase-10, they are able to make an interaction with DED of FADD, and thereby allow their association with the death receptor complexes [78]. The DED family proteins regulate apoptosis either by increasing caspase activation or preventing caspase activation by TNF family death receptors. One of the DED-containing proteins is FLIP, which is also known as FLAME, CLARP, CASH, CASPER, I-FLICE, MRIT or Usurpin [79]. It has been documented to be involved in suppressing apoptosis in cancer. The amino acid sequence of FLIP shares similarity with pro caspase-8 and -10 and, hence, competes with these caspases for binding to FADD, thereby silencing death receptor signaling. The increasing level of FLIP in tumor cells results in the development of resistance against apoptosis induction by Fas-expressing CTLs [80]. This FLIP-associated Fas resistance makes the tumor cell tolerate FasL expression, utilizing this death ligand as a weapon to nearby normal cells, and to stimulate apoptosis of immune cells. However, there are strategies for down-regulating the expression of FLIP employing antisense technology and drugs that restore sensitivity of tumor cell lines toward apoptosis towards FasL [81].

### 4.3. CARD-Family Proteins

There are numerous pro-caspases, namely, caspase-1, -2, -4, -5 and -9, which have CARDs in their N-terminal prodomains. The CARD has six helices present in DED and DD. CARD-family proteins are crucial in caspase activation via homotypic interactions throughout animal evolution. An example of caspase activation can be seen in CED-4 in *C. elegans* and APAF-1 in humans [82]. Both the proteins carry a CARD domain along with a nucleotide-binding oligomerization domain, called the NB-ARC (Nucleotide-binding domain homologous to APAF-1, CED-4 and plant R gene products) [83]. The CARD of APAF-1 associates with the CARD of pro-caspase-9, and upon oligomerization, they activate caspases. APAF-1 contains several WD-40 regulatory domains, which makes it dependent on cytochrome c. It has been seen that the oligomerization of APAF-1 is dependent upon dATP and cytochrome c, and once it is oligomerized, it binds and activates caspase-9 [82]. Cancer cells have shown innumerable mechanisms for preventing caspase activation via APAF-1, such as the inhibition of APAF-1 gene by methylation, increased expression of heat shock proteins that inhibit their functioning, increased expression of the tumor-up-regulated CARD-containing antagonist of caspase nine (TUCAN), which contains CARDs and is an antagonist of caspase-9, phosphorylating caspase-9 causing inhibition of their activity, and the association of CARDs with IAP-family proteins [73].

### 4.4. Inhibitor of Apoptosis Proteins

The inhibitor of apoptosis proteins or IAPs constitutes a family of suppressors of apoptosis that are conserved throughout the evolution. The main function of these proteins is the inhibition of caspases endogenously [84]. All IAPs have a common baculovirus inhibitor of the apoptosis protein repeat (BIR) domain, which is crucial for inhibiting apoptosis. However, only the presence of the BIR domain does not indicate anti-apoptotic activity as this domain is involved in regulating cell cycle with no impact on cell death [84]. Apart from the BIR domain, the IAP family of proteins also consists of other domains, such as RING zinc-fingers, CARDs, Ubiquitin-conjugating enzyme (E2s) domains and putative nucleotide-binding domains [85]. The RINGs and Apollon (BIR domain protein) are associated with ubiquitination machinery. Several IAPs, such as the X-linked inhibitor of the apoptosis protein (XIAP), cIAP1 and cIAP2 bind directly to the initiator caspase-9 and even to the executioner caspase-3 and caspase-7, thereby inhibiting their function. Different domains are required for inhibiting caspases; for instance, in the case of XIAP, a second BIR domain is required to inhibit caspase-3 and -7, while a third BIR domain is crucial for suppressing caspase-9 [86]. The IAP family members, namely, Livin and survivin, have

one BIR domain and suppress caspase-9, but not caspase-3 and -7. The IAPs are highly selective towards specific caspases; therefore, overexpression of IAPs can inhibit some of the apoptotic pathways but not all [87,88]. However, baculovirus p35 protein represents a broad-spectrum activity against most of the caspases, but no cellular homologue of this protein has yet been found. Further, IAPs target apoptosis mediated by intrinsic or extrinsic pathways, because their target, i.e., effector caspases are associated with these two pathways. It has been documented that overexpression of IAP family members is associated with cancers [89]. For instance, survivin, Livin, XIAP and cIAP1 overexpression are shown in melanomas and tumor cells [87,88]. However, antisense-mediated targeting of XIAP or cIAP1 can stimulate apoptosis in tumor cell lines. There are endogenous antagonists of IAPs, such as SMAC (DIABLO) and HTRA2 (OMI), which function in promoting apoptosis. These antagonists compete with caspases to bind to IAPs, thereby not allowing caspases from binding to IAP and, hence, favoring apoptosis [90].

*4.5. BCL-2 Family Proteins*

The BCL-2-family proteins are associated with the mitochondrial dependent pathway of apoptosis. Some proteins of this family, such as BCL-2, BCL-X$_L$ and BAK have a patch of hydrophobic amino acids at the C-terminal end through which these are linked with the outer mitochondrial membrane [91]. However, this hydrophobic patch is absent in BID, BIM and BAD, but they are associated with mitochondria via specific stimuli [92]. The BCL-2-family proteins are found to be conserved in the evolution of metazoan, and their homologues are present in vertebrates as well as invertebrates. Innumerable animal virus genomes, such as herpes simplex virus, Epstein-Barr virus (EBV) and Kaposi sarcoma herpes virus contain BCL-2 homologs [93]. There are 26 members of the BCL-2 family that are known currently. Their genes code for anti-apoptotic and pro-apoptotic proteins are, namely, BCL-2, BFL-1 (A1), BCL-X$_L$, MCL-1, BCL-w, BCL-B and BAX, BOK (MTD), BAK, BAD, BIM, BID, BIK, BCL-X$_S$, NIP3, HRK, PUMA, APR (NOXA), BCL-Gs, p193, NIX (BNIP) and BCL-RAMBO (MIL) [91,94–96]. The BCL-2 family members with their location and functions are summarized in Table 2. Few of the BCL-2 family genes generate more than two proteins via alternative splicing, which shows contrasting effects on apoptotic regulation, such as BCL-X$_L$ versus BCL-X$_S$ [97].

**Table 2.** The BCL-2 family proteins with their location and functions *.

| BCL-2 Protein | Location | Roles | Refs |
|---|---|---|---|
| BAX | Cytosol | Liberation of apoptogenic factors and induction of caspases | [98] |
| BAK | Integral mitochondrial membrane protein | Conformational changes in BAK take place to form larger complexes in apoptosis and create pores in the mitochondrial membrane to liberate apoptogenic factors to promote apoptosis | [99] |
| BID | Cytosol and membrane | Directly activate BAX | [100] |
| BCL-2 | Mitochondria, nucleus, endoplasmic reticulum | Prevents apoptosis by maintaining integrity of mitochondrial membrane integrity | [101,102] |
| BCL-X$_L$ | Mitochondrial transmembrane | Prevents release of cytochrome c via mitochondrial pore, thereby inhibiting activation of caspases by cytochrome c | [102] |
| MCL-1 | Nucleus, mitochondria | Associated with BAK1, BCL-2-associated death promoter, NOXA, BCL2L11 and PCNA | [103,104] |
| BCL-w/BCL2L2 | Mitochondrion | Under cytotoxic conditions downregulate apoptosis | [105] |

**Table 2.** *Cont.*

| BCL-2 Protein | Location | Roles | Refs |
|---|---|---|---|
| A1/BFL-1 | Mitochondria, nucleus | unknown | [106,107] |
| BIM/BCL2L11 | Mitochondria | Interacts with BCL-2 or BCL-$X_L$ and prevents their anti-apoptotic actions | [108] |
| PUMA | Mitochondria | unknown; regulated by p53 transcriptionally | [109,110] |
| BAD | Mitochondria | Generate a complex with BCL-2 and BCL-$X_L$, inhibits them, thereby promoting BAX/BAK-mediated apoptosis | [111] |
| BIK/BLK | Endoplasmic reticulum | unknown | [112] |
| NOXA/PMAIP1 | Mitochondria | unknown | [47,113] |
| BMF | Mitochondria | unknown | [114] |

\* Abbreviations used are: A1/BFL-1, BCL-2-related protein A1; BAD, BCL-2-associated agonist of cell death; BAK, BCL-2 antagonist killer; BAX, BCL-2-associated protein X; BCL-2, B-cell lymphoma 2; BCL-w/BCL2L2, BCL-2-like 2; BCL-$X_L$, B-cell lymphoma-extra-large; BID, BH3-interacting domain death agonist; BIK/BLK, BCL-2-interacting killer; BIM/BCL2L11, BCL-2-interacting protein BIM; BMF, BCL-2-modifying factor; MCL-1, myeloid cell leukemia 1; NOXA/ PMAIP1, phorbol-12-myristate-13acetate-induced protein 1; PUMA, p53-upregulated modulator of apoptosis.

Altogether, this section highlights the importance of DDs, CARDs, DEDs, BCL-2 family proteins and of IAP-family proteins in the regulation of apoptosis, and their possible link with lymphoproliferative syndrome, autoimmune syndrome and other malignancies. The next section focuses on the significance of apoptosis in carcinogenesis.

## 5. Apoptosis and Carcinogenesis

In multicellular organisms, there is a balance between cell formation via mitosis and cell death by apoptosis. Any imbalance in these regulatory processes leads to the development of cancer. Apoptosis is responsible for cell death, but dysfunction of it can cause malignancies [115]. In the early 1970s, Kerr and colleagues explained the link of apoptosis with elimination of malignant cells, regress tumor progression and hyperplasia [116]. It can be said that decline in apoptosis and resistance towards apoptosis plays a crucial role in cancer. There are several factors through which malignant cells can escape apoptosis or develop resistance against them. Some of the factors include upregulation in expression of IAPs, decline in expression of caspases, defects or mutations in p53, and misbalanced receptor signaling pathways and BCL-2 family of proteins, as shown in Figure 4 [117].

Some of the important strategies for targeting apoptosis for the treatment of cancer are summarized in Table 3. There are several treatment strategies or drugs that can act on apoptotic signaling pathways for the treatment of cancer and are mentioned in Table 3. Over the past three decades, scientists have been developing therapies using apoptosis for eliminating cancer cells [118,119]. These include the clinical translation of various pro-apoptotic members for drug discovery and also for understanding the cancer biology [115,120–122]. Collectively, these findings reveal that apoptosis plays an important role in the cytotoxicity in malignant tumors and could lead to the development of novel therapeutic techniques that target this process to regulate cancer cell proliferation.

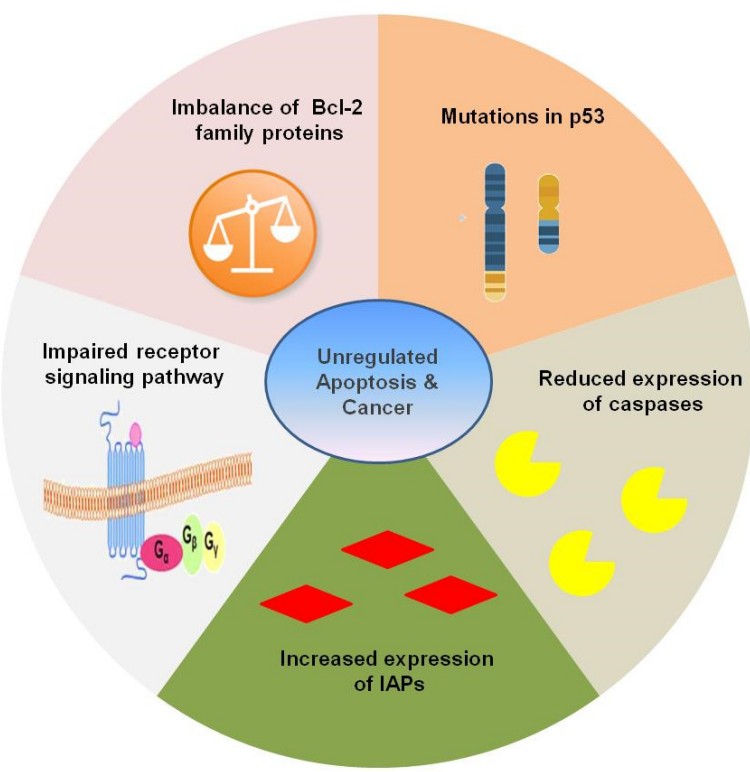

**Figure 4.** Some of the important factors in apoptotic signaling that are dysregulated during cancer.

**Table 3.** Targeting apoptosis and associated proteins for the treatment of cancer.

| Treatment | Remarks | Refs |
|---|---|---|
| | Attacking the BCL-2 family | |
| Oblimersen sodium | Showed chemosensitivity along with anticancer drugs with significant improvement in myeloid leukemia | [123,124] |
| BCL-2 family inhibitors (Small molecule) | Sodium butyrate, fenretinide, depsipetide and flavipirodo are known to alter gene or protein expression. While ABT-263, GX15-070, ABT-737, HA14-1 and gossypol, affect the proteins directly | [125] |
| BH3 mimetics | ABT-737 inhibit anti-apoptotic proteins namely BCL-2, BCL-$X_L$ and BCL-w | [126] |
| | ATF4, ATF3 and NOXA prevent MCL-1 functioning | [127] |
| Suppressing the Bcl family anti-apoptotic proteins/genes | BCL-2 specific siRNA prevent target gene expression and promote anti-proliferation and pro-apoptotic activity in pancreatic cancer cells | [128] |
| | Suppressing BMI-1 is known to decrease the expression of pAKT and BCL-2, it makes them sensitive to doxorubicin | [129] |
| | Targeting p53 | |
| p53-based gene therapy | Wild-type p53 genes having retroviral vector introduced into cancer cells showed significant improvement | [130] |
| | Introduction of wild type p53 gene makes head and neck tumor cells, and prostate cancers sensitive to radiotherapy | [131] |
| | ONYX-015 can disrupt tumor cells deficient in p53 | [132] |

**Table 3.** *Cont.*

| Treatment | Remarks | Refs |
|---|---|---|
| | Attacking the BCL-2 family | |
| | p53-dependent drug therapy | |
| Small molecules | PhiKan083 (A24275) binds and restores mutant p53 | [133] |
| | CP-31398 inserted with DNA and disrupts the DNA-p53 complex, leading to restoration of unstable p53 mutants | [134] |
| Other agents | Nutlins disrupt MSM2-p53 interaction, provide stability to p53 and promote death in cancer cells | [135] |
| | MI-219 breaks MDM2-p53 interaction, leading to inhibition of cell multiplication and the promotion of apoptosis in cancer cells | [136] |
| | Tenovins reduce tumor growth in vivo | [137] |
| p53-based immunotherapy | Vaccine having recombinant replication-defective adenoviral vector in combination with human wild-type p53 showed improvement | [138] |
| | p53-specific T cell responses seen when given p53 peptide | [139] |
| | Targeting inhibitors of apoptosis proteins (IAPs) | |
| Targeting XIAP by antisense approach | Improved tumor control by radiotherapy | [140] |
| | Antisense oligonucleotides increase chemotherapeutic activity | [141] |
| Targeting XIAP by siRNA approach | siRNAs targeting XIAP promote enhanced sensitivity towards radiotherapy | [142] |
| | siRNAs targeting XIAP make hepatoma cells sensitive towards death receptor and chemotherapy | [143] |
| Targeting survivin by antisense approach | Transfection of anti-sense survivin into melanoma cells promotes apoptosis | [144] |
| | Promote apoptosis and sensitivity of cancer cells towards chemotherapy | [145] |
| | Prevent growth of thyroid carcinoma cells | [146] |
| Targeting survivin by siRNA approach | Decrease survivin expression and lower the resistance to radiotherapy in pancreatic cancer cells | [147] |
| | Prevent proliferation and promote apoptosis in lung adenocarcinoma cells | [148] |
| | Downregulate survivin expression, prevent multiplication and increase apoptosis in ovarian cancer | [149] |
| | Increase radiosensitivity in cancer cells | [150] |
| IAP antagonists (Small molecules antagonists) | Hsp90 inhibitors and Cyclin-dependent kinase inhibitors are reported to target survivin | [151] |
| | Cyclopeptide SMAC mimetics 2 and 3 attaches to XIAP and cIAP-1/2, thereby promoting the induction of caspases- 9 and -3/-7 | [152] |
| | SM-164 increases TRAIL functioning | [153] |
| | Targeting caspases | |

**Table 3.** *Cont.*

| Treatment | Remarks | Refs |
|---|---|---|
| Caspase-dependent drug therapy | Apoptin promotes apoptosis in malignant cells | [154] |
| | Small molecule caspase activators stimulate caspase, promoting enhanced drug sensitivity in tumor cells | [155] |
| Caspase-dependent gene therapy | Caspase-3 gene therapy is reported to promote induction of extensive apoptosis | [156] |
| | Caspase-3 gene introduction into Huh7 human hepatoma cells promotes apoptosis | [157] |
| | Immunocaspase-3 in a recombinant adenovirus showed anticancer effect in hepatocellular cancer | [158] |

## 6. Targeting Apoptosis

### 6.1. Approaches Targeting Intrinsic Pathway of Apoptosis

The *BCL*-2 gene family can be targeted to control cancer growth as it has been seen that this gene is found in follicular lymphoma patients and can be targeted by promoting apoptosis. The approaches include small molecule mimetic of BH3 and small molecule inhibitors or oligonucleotides targeting *BCL-2* expression [121,159]. There are several ways to target intrinsic pathway of apoptosis (Table 4).

**Table 4.** Strategies to target intrinsic pathway of apoptosis *.

| Target | | Clinical Trial | Histology | Trial Identity ** |
|---|---|---|---|---|
| Dual BCL-2 and BCL-X$_L$ inhibitors | Navitoclax | YES (Phase I/II) | CLL, melanoma, solid tumors | NCT02079740, NCT02143401, NCT01989585, NCT02520778 |
| | APG-1252 | YES (Phase I/II) | SCLC, solid tumors | NCT03387332 |
| | AZD4320 | No | Childhood ALL | - |
| | S44563 | No | Melanoma, SCLC, | - |
| | BCL2–32 | No | NHL | - |
| | BM-1197 | No | Colorectal cancer | - |
| Selective BCL-2 inhibitors | Venetoclax | Yes (Phase I-III) | CLL, AML | - |
| | S55746 (BCL201) | Yes (Phase I) | NHL, multiple myeloma | NCT02603445, NCT02920697 |
| | APG-2575 | Yes (Phase I) | NHL, AML | NCT03537482, NCT03913949 |
| BCL-X$_L$ inhibitors | ABBV-155 * | Yes (Phase I) | Solid tumors | NCT03595059 |
| | WEHI-539 | No | Breast cancer | - |
| | A-1155463 | No | AML | - |
| | A-1331852 | No | Soft-tissue sarcoma | - |

**Table 4.** *Cont.*

| Target | | Clinical Trial | Histology | Trial Identity ** |
|---|---|---|---|---|
| MCL-1 inhibitors | AMG 176 | Yes (Phase I) | NHL, AML | NCT02675452, NCT03797261 |
| | MIK665 (S64315) | Yes (Phase I) | NHL, AML | NCT02992483, NCT03672695, NCT02979366 |
| | AZD5991 | Yes (Phase I) | NHL, AML | NCT03218683 |
| | S63845 | No | NHL, AML | - |
| | UMI-77 | No | Pancreatic cancer | - |
| | A-1210477 | No | Esophageal carcinoma | - |
| | VU661013 | No | AML | - |
| IAP inhibitors and SMAC mimetic antagonists | LCL161 | Yes (Phase I/II) | Colorectal cancer, multiple myeloma, Polycythemia vera, myelofibrosis | NCT02649673, NCT02098161, NCT03111992 |
| | Birinapant (TL32711) | Yes (Phase I/II) | Advanced solid tumors, NHL | NCT03803774, NCT02587962 |

* Abbreviations used are: ALL, acute lymphoblastic leukemia; AML, acute myelogenous leukemia; BCL-2, B-cell lymphoma 2; BCL-$X_L$, B-cell lymphoma-extra-large; CLL, chronic lymphocytic leukemia; IAP, inhibitor of apoptosis; MCL-1, myeloid cell leukemia 1; NHL, Non-Hodgkin lymphoma; SCLC, small cell lung cancer; SMAC, small mitochondrial-derived activator of caspase. ** Clinical trial numbers were taken from ClincalTrials.gov [160].

6.1.1. BH3 Mimetics

The first inhibitor molecule designed against BCL-2, BCL-$X_L$ and BCL-w was ABT-737. This molecule binds in the hydrophobic pocket of BCL-2 family members, where BH3-only protein binds. It shows good results against lung carcinoma and worked in combination with chemotherapy and radiation [161]. This is followed by the discovery of navitoclax (ABT-263), which also showed anti-cancer therapeutic potential, and in synergy with MEK or tyrosine kinase inhibitors it worked against solid tumors [162]. Moreover, venetoclax (ABT-199) is a BCL-2 small molecule inhibitor that exhibits significant results against CLL and non-Hodgkin lymphoma (NHL), which show an increased expression of BCL-2 [163]. Additionally, selective BCL-$X_L$ inhibitors are also in the development phase, such as a BCL-$X_L$-based vaccine for prostate cancer and the inhibitor of BCL-$X_L$, i.e., an antibody–drug conjugate, namely, ABBV-155, which focuses on solid tumors as a monotherapy or along with taxanes. Interestingly, BH3 mimetics have been developed successfully due to the advent in technologies for targeting protein–protein interactions utilizing stapled peptides, i.e., the synthetic protein becomes trapped within the secondary structure via a chemical staple. These peptides directly target the protein–protein interaction with enhanced penetration inside the cell [164]. The first example of a staple peptide is SAHB$_A$ (stabilized alpha-helix of BCL-2 domains), one which mimics the $\alpha$-helical BH3 segment of proapoptotic BID. It easily penetrates inside the leukemia cells and becomes attached to BCL-$X_L$, stimulates apoptosis and shows anti-tumor activity [165].

Additionally, stapled peptides work against the p53-MDM2 interaction for reactivating p53 and similar compounds as well as targeting anti-apoptotic BFL-1 and MCL-1 [166]. In lung cancer, apoptosis can be induced by targeting the regulatory site, i.e., S184 of Bax, which controls its localization. This site can be targeted via a small molecule, namely, SMBA1-3, which specifically binds to Bax and prevents the phosphorylation of S184, thereby promoting the liberation of cytochrome c and finally leading to apoptosis [167]. Moreover, some other Bax-activating molecules, such as BAM-7 and BTSA1, also showed significant results against glioblastoma cell lines, while small molecule inhibitors of Bax, known as Bax activation inhibitors, can also show their allosteric inhibition with significant antitumor potential.

### 6.1.2. MCL-1 Inhibitors

MCL-1 is associated with several forms of malignancies and is involved in resistance to chemotherapy and BCL-2 and BCL-X$_L$ inhibitors [168]. A small molecule inhibitor, namely, AM-8621, has the ability to bind to the MCL-1 pocket and remove BIM and promote apoptosis in myeloma cell lines. Additionally, the derivatives of this small molecule, i.e., AMG 176 and AZD5991, have also shown significant results in good synergy with venetoclax and chemotherapy [169,170]. Moreover, the MCL-1 inhibitors, VU661013 and S63845, showed good results in hematological malignancies and in combination strategies such as to combat venetoclax resistance [121].

### 6.1.3. IAP Inhibitors

IAP inhibitors are considered as a potential tool to promote apoptosis in cancer cells. There are eight known IAPs in humans, and among these, the most significant with anti-apoptotic activity are XIAP, melanoma IAP (ML-IAP), and cellular IAP1 and IAP2 [171]. Interestingly, SMAC and OMI/HTRA2 have the ability to specifically and independently inhibit XIAP. SMAC itself binds to XIAP, c-IAP1 and c-IAP2, and thereby inhibits XIAP from binding caspase-3, -7 and -9. IAP proteins play an important role in cell survival regulation; hence, inhibitors of IAP and SMAC mimetics could be of great therapeutic application especially in cancers [172]. IAP antagonists such as LCL161 showed anti-tumor potential and in combination with chemotherapy also displayed good results in cases of multiple myeloma and head and neck squamous carcinoma models. One more compound, birinapant (TL32711), an IAP antagonist, did not display good activity singly, but when combined with radiotherapy and anti-PD1 pembrolizumab, it showed significant results in head and neck cancer [173].

### *6.2. Approaches Targeting Extrinsic Pathway*

### 6.2.1. Death Receptor Agonists

The extrinsic pathway of apoptosis is stimulated by extracellular signals that activate transmembrane protein members of the TNF receptor superfamily (TNFR) and pro-apoptotic death receptors (DRs). Death receptors, namely, TNFR1, Fas (CD95, APO-1), DR3, DR4 (TRAILR1), DR5 (TRAILR2) and DR6, once bound to their specific ligands, i.e., Fas, TNFR1, TNFR2, DR4 and DR5, allow binding of an adapter protein such as FADD, leading to the generation of the DISC, which stimulates caspase-8 and -10 and the downstream signaling of apoptotic pathway [174]. Therapeutic strategies are developed for targeting DR4 and DR5 by developing recombinant Apo2L/TRAIL (stimulating DR4 and DR5) as well as agonistic monoclonal antibodies targeting DR4 or DR5 [175,176]. Dulanermin, a recombinant APO2L/TRAIL, was utilized in clinical trials along with other drugs and chemotherapy in cases of solid tumors, but these did not show very promising results due to its short half-life [177]. Agonist antibodies were also developed against DR4 and DR5 as they have a good half-life and they showed significant preclinical results. Agonist monoclonal antibodies, namely, mapatumumab and lexatumumab for DR4 and DR5, respectively, were well tolerated in clinical trials and were administered singly as well in combination with other drugs and chemotherapy in solid tumors adult sarcomas [178,179]. Several other DR5 agonists were also tried in clinical trials, such as conatumumab (AMG655), tigatuzumab, LBY135 and drozitumab (PRO95780) [180,181]. Furthermore, TRAIL-inducing small molecules were also developed as the expression of TRAIL is associated with anti-tumor activity. One such TRAIL-inducing small molecule is ONC201 [182]. It binds to dopamine receptors, namely, DRD2 and DRD3, and can also bind to mitochondrial caseinolytic protease P (CIpP). Once this binding takes place, the integrated stress response protein ATF4 becomes activated, which leads to the upregulation of DR5 and finally apoptosis [182–184].

### 6.2.2. Tumor Suppressor Pathways

There are several therapeutic approaches targeting tumor suppressor pathways that are being developed as promising anti-tumor agents. Inhibition of unregulated oncogenic effectors, such as PI3K, AKT, β-catenin, Myc, CDKs, mTOR and VEGF, on mutated tumor suppressors is under exploration, showing significant clinical outcomes [185]. A large number of strategies targeting the p53 pathway are under development as this is one of the major targets to be inactivated in case of cancers [186]. Moreover, MDM2 and MDMX inhibitors are under investigation as they can block the pathway of wild-type p53 degradation without damaging DNA. MDM2 inhibitors, such as nutlin-3a derivative RG7388 (idasanutlin), are administered either singly or in combination with venetoclax, atezolizumab or chemotherapy [187]. Some other MDM2 antagonists under investigation include AMG-232, HDM201, APG-115, DS-3032b, BI 907828 and ALRN-6924. Trials are also being conducted for combining MDM2 antagonists along with inhibitors of other targeted agents such as MEK 1/2, BCL-2/BCL-X$_L$, PI3K and BRAF inhibitors [121]. Moreover, strategies targeting restoration of the p53 tumor suppressor pathway in tumors having a p53 mutant can be explained by APR-246. The restoration of the p53 pathway in p53-mutated tumors promotes apoptosis in tumors [188]. Additionally, inhibitors of CDK4/6, such as palbociclib, are approved for breast cancer as they promote cell death by increasing the cell sensitivity towards TRAIL-mediated apoptosis. They can also promote cell cycle arrest in glioblastoma and myeloma [189].

### 6.2.3. Epigenetic Approaches Targeting Apoptosis

Histone deacetylases and BET are epigenetic modulators and their inhibitors possess anticancer activity via apoptosis and have shown good synergy with BCL-2 inhibitors [190,191]. A BET inhibitor, namely, ABBV-075, in combination with venetoclax has shown good results in patients having cutaneous T cell lymphoma (CTCL) [191]. Its treatment has shown a decline in the protein expression of MCL-1, BCL-2 and BCL-X$_L$ in the acute myelogenous leukemia (AML) cell lines due to BET inhibitor-mediated modulation of gene expression and chromatin.

Furthermore, the inhibitors of HDAC affect chromatin remodulation and promote apoptosis by different processes. They increase expression of death receptors, TRAIL, Fas ligands, and BH3-only proteins such as BID, BMF, BIM and BAK. In addition, they reduce expression of BCL-2, BCL-w, BCL-X$_L$ and MCL-1, with an increment in the generation of reactive oxygen species, which in turn stimulates the intrinsic pathway of apoptosis [190]. HDAC inhibitors such as panobinostat stimulate NOXA and downregulate MCL-1 in B cell lymphoma cell lines and also increase sensitivity towards the BCL-2 inhibitor [192]. HDAC inhibitors can also increase the performance of MEK inhibitors and venetoclax in multiple myeloma. Moreover, a hypomethylating agent, namely, azacytidine, demonstrated good results with venetoclax and ABT-737 [193].

### 6.2.4. Chaperons Targeting Apoptosis

Several strategies that induce stress in cells and finally promote apoptosis are being explored to target cancer cells. Chaperones are members of the heat shock protein family make the proteins stable in cancer cells; hence, inhibitors of the heat shock protein, specifically hsp90, which stabilizes the protein, are being explored [194]. The inhibitors of hsp90, namely, geldanamycin, ganatespib, onalespib, XL888 and TAS116, are under investigation [195]. Moreover, the stress-mediated by the endoplasmic reticulum also promotes apoptosis in association with FADD, protein kinase R-like endoplasmic reticulum kinase (PERK), caspase-8 and enhanced expression of ER chaperone GRP78 [196]. Furthermore, the knowledge of ER-mediated stress, apoptosis and chaperons in cancer pathogenesis should be explored to a greater extent, and inhibitors targeting chaperons that stabilize the protein in cancerous cells should be targeted [121]. Table 5 enlists the strategies used for targeting extrinsic pathways. Finally, the findings from the different trials have reignited interest in targeting apoptosis pathways. In particular, interfering with the function of

specific proteins, modulation of epigenetic modulations or alterations, as well as the of chaperon activity, are promising strategies for the generation of therapeutic strategies in oncology.

**Table 5.** Strategies to target extrinsic pathway and other associated death mechanism of apoptosis.

| Target | Clinical Trials | Cancer | Trial Identity * |
|---|---|---|---|
| Death Receptor Agonists (DR4/5) | | | |
| GEN1029 | Yes (Phase I) | Colorectal cancer, renal carcinoma, triple negative breast cancer, pancreatic cancer, gastric cancer | NCT03576131 |
| ABBV-621 | Yes (Phase I) | AML, NHL, pancreatic cancer | NCT03082209 |
| MM-201 | No | Sarcoma | - |
| TLY012 | No | Fibrosis | - |
| Approaches targeting p53 | | | |
| Idasanutlin (RG73882) | Yes (Phase I) | Breast cancer, AML, NHL, multiple myeloma | NCT03850535, NCT02670044, NCT02545283, NCT02633059, NCT03566485, NCT03135262 |
| AMG-232 | Yes (Phase I/II) | AML, multiple myeloma, sarcoma | NCT03041688, NCT03217266, NCT03031730 |
| HDM201 | Yes (Phase I) | AML | NCT03940352 |
| APG-115 | Yes (Phase I) | Advanced solid tumors, AML, melanoma | NCT02935907, NCT03611868, NCT03781986 |
| DS-3032b | Yes (Phase I) | AML, solid tumors | NCT03634228, NCT02319369, NCT01877382 |
| BI 907828 | Yes (Phase I) | Solid tumors | NCT03449381 |
| ALRN-6924 *** | Yes (Phase I) | Solid tumors | NCT03725436 |
| Restore wild type activity of mutant p53 | | | |
| APR246 | Yes (Phase I) | AML, esophageal carcinoma, ovarian cancer, melanoma | NCT02999893, NCT02098343, NCT03588078, NCT03391050, NCT03391050 |
| Other cell death mechanisms associated with apoptosis | | | |
| ONC201 ** | Yes (Phase I/II) | NHL, breast cancer, multiple myeloma, colorectal cancer, endometrial cancer, AML | NCT03099499, NCT02863991, NCT03416530, NCT02420795, NCT03394027, NCT03295396, NCT03791398, NCT02392572 |
| Epigenetic modulators for stimulating intrinsic pathway of apoptosis | | | |
| Fimepinostat +venetoclax | Yes (Phase I/II) | NHL | NCT01742988 |
| Azacytidine or decitabine + venetoclax | Yes (Phase I-III) | AML | NCT03404193, NCT03941964 |

Abbreviations used are: AML, acute myelogenous leukemia; NHL, Non-Hodgkin lymphoma. *** ALRN-6924 is the first-ever clinical stage stapled peptide, ** ONC201 binds to DRD2/DRD3 and CIpP resulting in activation of stress response protein ATF4 and cell death, * Clinical trial numbers were taken from ClincalTrials.gov [160].

## 7. Conclusions

Apoptosis is considered as an energy-dependent process, which is associated with several biochemical and morphological characteristics where caspase induction plays a crucial role. However, several other apoptotic proteins, pro-apoptotic and anti-apoptotic proteins, are also involved in the regulation of apoptosis. The understanding of the mechanism of apoptosis is crucial because it balances both cell survival and cell death. It has been suggested that any defects in the apoptotic pathway are involved in cancer and several approaches targeting apoptosis for treating cancer are feasible and are under investigation. Therefore, the exploration and knowledge of the mechanism of apoptosis, and associated proteins at the cellular and molecular level, would help in digging deeper insights into several disease processes caused by apoptosis and, thereby, aiding in developing therapeutic strategies.

There is some evidence that suggests that the resistance to apoptosis is one of the major causes of cancer. So, targeting apoptotic pathways for the treatment of cancer is therapeutically important. Therefore, both targeting of intrinsic and extrinsic pathways of apoptosis as well as interfering with other associated cell death mechanisms of apoptosis are enticing anti-cancer strategies. Currently, there are several drugs tested in clinical trials as a monotherapy, as well as in combination with chemotherapy, radiation therapy, or other inhibitors for the management of different cancers. However, it is very challenging to modulate apoptosis pathways using targeted agents as there are several pathways interfering with apoptosis and involved in the formation of cancer resistance. Hence, using rational combination methods could be the key to their clinical success to overcome these resistance mechanisms. Overall, we infer that targeting apoptosis mechanisms could be a promising oncology therapy that will continue to evolve in clinical practice in the future.

**Author Contributions:** Conceptualization, A.K.; resources, K.K.B. and R.W.; writing—original draft preparation, V.S., A.K., U.N., P.A. and R.W.; writing—review and editing, V.S., A.K., U.N., P.A., K.K.B. and R.W.; visualization, V.S., A.K. and P.A. All authors have read and agreed to the published version of the manuscript.

**Funding:** A.K. has received a P.R.I.M.E scholarship from the DAAD. R.W. is sponsored by the German Research Foundation (grants WE 2554/13-1, WE 2554/15-1, and WE 2554/17-1).

**Institutional Review Board Statement:** Not applicable.

**Informed Consent Statement:** Not applicable.

**Data Availability Statement:** Not applicable.

**Conflicts of Interest:** The authors declare no conflict of interest.

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
