# Peer review of "Apoptosis and Pharmacological Therapies for Targeting Thereof for Cancer Therapeutics"

_sci, doi:10.3390/sci4020015_

Round 1

Reviewer 1 Report

It is a very comprehensive review on apoptosis and its connection with cancer therapies, written in an interesting and accessible way.

A small very subjective remark: it is much easier to read a table if the text in each column is left-aligned.

Author Response

Dear reviewer,

please find attached a pdf-file in which we decribe how we responded to your valuable comments.

Reviewer 2 Report

The article "Apoptosis and Pharmacological Therapies for Targeting Thereof for Cancer Therapeutics" describes the process of apoptosis and its different pathways and different approaches of cancer treatment by targeting apoptosis-involved proteins. While the article is informative, it lacks any observations or comments on the matter from the authors. Also, English and style needs much improvement. Below I give some of the major and minor comments.

Major

1. English must be improved a little - line 19, 47, 68, 78-81 (few errors, e.g., “is evidence”, 86-87 (repeated “other”), 103, 108, 139, 166, 209, and many more. Also please avoid excessive repetitions of word “apoptosis” in one or two sentences throughout the manuscript like in lines 60-61. I highly recommend using professional English editing services, as many spelling mistakes can be found, and the flow of the text could be very much improved.

2. Please reread the text and correct/remove all vague phrases such as “like, certain, various, generally etc.” and give specific facts instead.

3. Why the perforin/granzyme and execution pathways are not shown/marked in the figure?

4. In my opinion such detailed description of caspases is not necessary. The majority of information given in section 3 could be presented and compared in a table for better clarity or the table 1 could be improved for additional information which would allow to shorten the section and remove all information already given in table.

5. Tables are the best part of the manuscript; however, I believe they should be merged into one and unified as they all represent ways to target apoptosis for cancer treatment. One table and short paragraphs describing particular groups of possible treatments would be, in my opinion, easier to follow and more beneficial for the reader.

6. The manuscript lacks some insights or thought form authors. It only describes given approaches. There should be a comment (possibly at the end of each section/subsection) on which approach is possibly better, gives better outlooks. The article would greatly benefit from adding authors’ observations and thought.

Minor

1. line 43 – please give example of other ways of programmed cell death

2. Please explain all used abbreviations at the first use e.g., Bcl-2 in abstract, BH3, Bid in Fig1 caption, all of the used in Fig2 etc. All abbreviations used in figures should be explained in the figures’ captions. Correct and add where appropriate.

3. Refrain from using “like” (e.g., lines 69,70), it is not quite professional language.

4. Line 69 – “like corticosteroids” – give examples. Also, this sentence indicates that hormones follow the path of apoptosis. Please correct.

5. Line 70 – “certain circumstances” – describe those circumstances.

Author Response

(The authors gave the same response as above.)

Reviewer 3 Report

The authors have compiled information on apoptotic signaling pathways and mechanisms involved and associated with carcinogenesis and therapeutics. The manuscript is somewhat nicely written and covers a good range of information regarding apoptosis and cancer therapeutics. The science structure and overall concept of the manuscript are sound and acceptable. However, I have some queries that need to be addressed.

Comment: The major issue with the write-up is that it lacks novelty; authors must explain how it’s different from the sea of information already available on the topic.

Comment: There is no specific concluding information in the introduction section which justifies why this review is essential.

Comment: Provide more discussion/analysis by the authors rather than a summary of literature information.

Comment: Table 1 compiles the role of caspases with their location, substrate, and functions lacks proper referencing. In a similar way as in table 2

Comment: The authors should carefully re-organize the structure, polish their language, avoid self-repetition, and eliminate simple errors.

Comment: Authors must check reference style as there are many mistakes in font and style. There are only three references from the last three years; authors are advised to add more Also, cite the following important references in the text.

Shoaib A, Tabish M, Ali S, Arafah A, Wahab S, Almarshad FM, Rashid S, Rehman MU. Dietary Phytochemicals in Cancer Signalling Pathways: Role of miRNA Targeting. Curr Med Chem. 2021;28(39):8036-8067.

Carneiro BA, El-Deiry WS. Targeting apoptosis in cancer therapy. Nat Rev Clin Oncol. 2020 Jul;17(7):395-417.

Wani JA, Majid S, Khan A, Arafah A, Ahmad A, Jan BL, Shah NN, Kazi M, Rehman MU. Clinico-Pathological Importance of miR-146a in Lung Cancer. Diagnostics (Basel). 2021 Feb 10;11(2):274.

Neophytou CM, Trougakos IP, Erin N, Papageorgis P. Apoptosis Deregulation and the Development of Cancer Multi-Drug Resistance. Cancers (Basel). 2021 Aug 28;13(17):4363.

Rehman MU, Khan A, Imtiyaz Z, Ali S, Makeen HA, Rashid S, Arafah A. Current nano-therapeutic approaches ameliorating inflammation in cancer progression. Semin Cancer Biol. 2022 Feb 7:S1044-579X(22)00027-X.

Author Response

(The authors gave the same response as above.)

Round 2

Reviewer 2 Report

The article "Apoptosis and Pharmacological Therapies for Targeting Thereof for Cancer Therapeutics" has been improved by the authors, however, in my opinion, the most important issue is still lacking. Please see the comment below.

  1. The manuscript lacks some insights or thoughts from the authors. It only describes given approaches. There should be a comment (possibly at the end of each section/subsection) on which approach is possibly better, gives better outlooks. The article would greatly benefit from adding the authors’ observations and thoughts.

We fully agree with your comment. In the revised version, we have added conclusive comments at the end of each section.

Response: The comments added by the authors at the end of each section are solely a simple summary of each chapter and they only describe what was presented in individual chapters. The authors did not provide any insights or thoughts of their own. The commentary on described approaches is still lacking e.g., giving outlooks, possible application, commenting on which approach is possibly better for particular situations, referring to and discussing current developments in the field, etc.

Lines 183-185 are unnecessary, as it does not tell the reader why the different pathways are important, it only repeats what was described.

Lines 386-370 state the obvious and do not provide any insights.

Conclusions would be the place to put the article in a broader context and summarize the authors' unique observations on the topic, not the only state that the “The understanding of the mechanism of apoptosis is crucial […]” and “Therefore, the exploration and knowledge of the mechanism of apoptosis, and associated proteins at the cellular and molecular level would help in digging deeper insights into several disease processes […]”, as this, again, states the obvious, and does not introduce any interest to the reader.

Author Response

Dear reviewer,

again many thanks for reviewing our paper. Please find our comments in the attached pdf-File.

Regards

Ralf Weiskirchen

Reviewer 3 Report

The authors have revised the manuscript as suggested. No further changes are needed.

Author Response

Dear reviewer,

many thanks for accepting our paper.

Regards

Ralf Weiskirchen

Round 3

Reviewer 2 Report

The authors slightly improved the manuscript according to the previous comments.